# Bat Rhinacoviruses Related to Swine Acute Diarrhoea Syndrome Coronavirus Evolve under Strong Host and Geographic Constraints in China and Vietnam

**DOI:** 10.3390/v16071114

**Published:** 2024-07-11

**Authors:** Alexandre Hassanin, Vuong Tan Tu, Phu Van Pham, Lam Quang Ngon, Thanina Chabane, Laurent Moulin, Sébastien Wurtzer

**Affiliations:** 1Institut de Systématique, Évolution, Biodiversité (ISYEB), SU, MNHN, CNRS, EPHE, UA, Sorbonne Université, 75005 Paris, France; thanina.chabane@etu.u-paris.fr; 2Institute of Ecology and Biological Resources, Vietnam Academy of Science and Technology, No. 18, Hoang Quoc Viet Road, Cau Giay District, Hanoi 10072, Vietnam; tuvuongtan@gmail.com (V.T.T.); phupham.iebr@gmail.com (P.V.P.); ngoneco@gmail.com (L.Q.N.); 3Eau de Paris, R&D Laboratory, Direction Recherche, Développement et Qualité de l’Eau, 94200 Ivry-sur-Seine, France; laurent.moulin@eaudeparis.fr (L.M.); sebastien.wurtzer@eaudeparis.fr (S.W.)

**Keywords:** RNA virus, coronavirus, *Alphacoronavirus*, SADS-CoV, recombination, consensus sequence, supertree, phylogeography, Southeast Asia, genomic bootstrap barcode

## Abstract

Swine acute diarrhoea syndrome coronavirus (SADS-CoV; Coronaviridae, *Rhinacovirus*) was detected in 2017 in Guangdong Province (China), where it caused high mortality rates in piglets. According to previous studies, SADS-CoV evolved from horseshoe bat reservoirs. Here, we report the first five *Rhinacovirus* genomes sequenced in horseshoe bats from Vietnam and their comparisons with data published in China. Our phylogenetic analyses provided evidence for four groups: rhinacoviruses from *Rhinolphus pusillus* bats, including one from Vietnam; bat rhinacoviruses from Hainan; bat rhinacoviruses from Yunnan showing a divergent synonymous nucleotide composition; and SADS-CoV and related bat viruses, including four rhinacoviruses from Vietnam sampled in *Rhinolophus affinis* and *Rhinolophus thomasi*. Our phylogeographic analyses showed that bat rhinacoviruses from Dien Bien (Vietnam) share more affinities with those from Yunnan (China) and that the ancestor of SADS-CoVs arose in *Rhinolophus affinis* circulating in Guangdong. We detected sequencing errors and artificial chimeric genomes in published data. The two SADS-CoV genomes previously identified as recombinant could also be problematic. The reliable data currently available, therefore, suggests that all SADS-CoV strains originate from a single bat source and that the virus has been spreading in pig farms in several provinces of China for at least seven years since the first outbreak in August 2016.

## 1. Introduction

Severe acute diarrhoea syndrome coronavirus (SADS-CoV) was initially detected between January and May 2017 in several pig farms located in Guangdong Province (south-eastern China), where it was responsible for acute vomiting and watery diarrhoea in piglets, causing the death of about 25,000 animals [1,2,3]. The positive single-stranded RNA genome of SADS-CoV is 27,155–27,200 nucleotides (nt) long and contains nine open reading frames (ORFs): ORF1a and ORF1b that encode for a replicase–transcriptase composed of 16 non-structural proteins (NSP1–16); four ORFs that encode structural proteins, i.e., spike (S), envelope (E), membrane (M), and nucleocapsid (N); an accessory ORF3 between S and E genes, and two overlapping ORFs (NS7a and NS7b) following the N gene [1,2,3].

Subsequently, a retrospective study revealed that SADS-CoV had circulated in Guangdong as early as August 2016 [4]. In 2018, a new SADS-CoV strain, CH/FJWT/2018, was detected in diarrhoeal piglets from seven farms in Fujian Province, China; its genome showed 99.6% of identity with previous SADS-CoV strains and differed from them by a few insertions also found in several bat rhinacoviruses, suggesting that CH/FJWT/2018 “might originate from bats” [5]. In February 2019, SADS-CoV re-emerged in Guangdong, with an outbreak that caused the death of about 2000 pigs; the new SADS-CoV strain, CN/GDLX/2019, showed 99.3–100% of genome identity with SADS-CoVs reported previously [6]. In May 2021, SADS-CoV re-emerged in an intensive-scale pig farm in Guangxi Province, with an outbreak that caused the death of 3000 piglets; the new SADS-CoV strain, Guangxi/2021, exhibited only 98.4–98.8% of genome identity with other SADS-CoVs; it was described as a recombinant between two SADS-CoVs [7].

Phylogenetic studies based on alignments of whole coronavirus genomes have shown that all SADS-CoVs form a monophyletic group related to HKU2 viruses sampled from bats collected in China [3,7,8,9]. The current taxonomic classifications indicate that SADS-CoV and bat-related coronaviruses belong to the family Coronaviridae, genus *Alphacoronavirus*, and subgenus *Rhinacovirus* (https://ictv.global/, accessed on 1 May 2024; NCBI: txid2509509 at https://ncbi.nlm.nih.gov/taxonomy/, accessed on 1 May 2024). As for coronaviruses related to SARS-CoV and SARS-CoV-2 (genus *Betacoronavirus*, subgenus *Sarbecovirus*), there is strong evidence supporting that horseshoe bats (family Rhinolophidae, genus *Rhinolophus*) are the main reservoir host of rhinacoviruses [3,10], in which their genomes evolve through the combined actions of nucleotide mutation, inter-genomic recombination, and natural selection.

In a previous study, we sequenced and assembled 38 genomes of sarbecoviruses related to SARS-CoV and SARS-CoV-2 in faecal samples from horseshoe bats captured in several provinces of Vietnam [11]. Among the reads generated by metagenomic sequencing, we detected several coinfections with rhinacovirus. Here, we report the first complete *Rhinacovirus* genomes extracted from horseshoe bats circulating in three provinces of Vietnam, i.e., Cao Bang, Dien Bien, and Quang Tri. The five *Rhinacovirus* genomes were compared to those previously sequenced in China. Our three main goals were (i) to analyse the synonymous nucleotide composition of *Rhinacovirus* genomes, (ii) to better understand the geographic and host evolution of rhinacoviruses in horseshoe bats from China and Vietnam, and (iii) to further explore the recombination in SADS-CoV strains isolated from farmed pigs in southern China.

## 2. Materials and Methods

### 2.1. RNA Extraction, Sequencing, and Genome Assembly

The protocols used for bat sampling, RNA extraction, and sequencing were detailed previously [11]. In brief, bats were captured during several field surveys in Vietnam between October 2021 and November 2022, and 547 RNA extractions were done from faecal samples, providing 59 positive PCR amplifications of the Envelope (E) gene of *Sarbecovirus*. Most of these samples were then prepared with a Stranded Total RNA Prep with Ribo-Zero plus kit (Illumina, San Diego, CA, USA) and sequenced using a NovaSeq 6000 S1 Reagent kit (300 cycles) at the “Institut du Cerveau et de la Moelle épinière” (Paris, France). The Illumina reads were mapped in Geneious Prime^®^ 2020.0.3 using 20% of maximum mismatches per read on 63 coronavirus reference genomes representing a large diversity of viruses from the four genera *Alphacoronavirus*, *Betacoronavirus*, *Deltacoronavirus*, and *Gammacoronavirus* (family Coronaviridae, subfamily Orthocoronavirinae). In this way, we were able to detect five samples showing both *Sarbecovirus* and *Rhinacovirus* reads. Then, the first contigs were further extended using iteration mapping in Geneious Prime^®^ with 3% or 1% of maximum mismatches per read and aligned to each other in order to achieve a complete genome assembly.

### 2.2. Whole-Genome Alignment of Rhinacoviruses

All complete genomes of the subgenus *Rhinacovirus* available in GenBank were downloaded from https://www.ncbi.nlm.nih.gov/ (accessed on 1 December 2023). To be exhaustive, we conducted BLAST searches in NCBI using two reference genomes of *Rhinacovirus* (Coronaviridae; Orthocoronavirinae; *Alphacoronavirus*): NC_009988 (*Rhinolophus* bat coronavirus HKU2 [12]; code used hereafter: RsHKU2) and NC_028824 (BtRf-AlphaCoV/YN2012 [13]; code used hereafter: RfYN2012). With both genomes, we found a strong gap in BLAST results: all genomes of the subgenus *Rhinacovirus* showed more than 82% of nucleotide identity, whereas genomes of other subgenera of *Alphacoronavirus*, such as *Decacovirus*, *Minunacovirus,* and *Pedacovirus*, showed no more than 75% of nucleotide identity. The details on the 92 *Rhinacovirus* genomes selected for our study are provided in Appendix A. All of them were sampled in China, and for most of them, the authors indicated in GenBank the binomial name of the host species (91%) and the province where the sample was collected (51%). Based on the literature, we were able to complete the information for all genomes (Figure 1) except Ra160660, for which we could not find the province of origin in China (the authors did not respond to our request). In addition, only a few research teams provided the collection date and GPS coordinates of rhinacoviruses (34% and 2%, respectively; Figure 1), whereas these data are known to be essential for studies on molecular dating and phylogeography.

Our five *Rhinacovirus* genomes from Vietnam were compared to those previously sampled in China. We removed from our dataset two SADS-CoV genomes, i.e., GenBank accessions MK994936 and MK994937, because they involved many cell culture passages (48 and 83, respectively) [14]. In total, 95 nucleotide sequences were aligned in Geneious Prime^®^ 2020.0.3 with MAFFT 7.450 [15] using default parameters. Then, the alignment was corrected manually using AliView 1.26 [16] based on the three following criteria applied to nucleotide and amino acid sequences: (i) the number of indels was minimized because they are rarer events than nucleotide substitutions; (ii) transitions were privileged over transversions because they are more frequent; and (iii) changes between similar amino acids (as shown by the ClustalX colour scheme) were preferred. Our final alignment includes 95 *Rhinacovirus* genomes and 27,250 nucleotide sites (nt).

### 2.3. Analysis of Synonymous Nucleotide Composition (SNC)

The SNC of *Rhinacovirus* genomes was studied using an alignment reduced to all protein genes (26,682 nt). Nucleotide frequencies were calculated as detailed previously [17], and the variables were summarised by a principal component (PC) analysis using the FactoMineR package [18] in R version 3.6.1 (from https://www.R-project.org/, accessed on 1 December 2023).

### 2.4. Phylogenetic Analyses

Phylogenetic analyses were carried out on IQ-TREE version 2.2.2.6 [19] using the whole-genome alignment of 95 rhinacoviruses, the GTR+F+I+R5 model, and 1000 bootstrap replicates. We also performed a similar analysis based on a reduced alignment of the Spike gene (3435 nt).

To examine the distribution of phylogenetic support along the whole-genome alignment, the dataset was bootstrapped under the SWB (sliding window bootstrap) program [20] following the procedure detailed in [21]: five different SWB analyses were conducted using five window sizes (i.e., 400, 500, 600, 1000, and 2000 nt) moving in steps of 50 nt (Appendix A). In the SWB program, each window sub-dataset was automatically run in RAxML [22] with a GTR+G model and 100 bootstrap replicates. The SWB output file contains the window bootstrap percentages (WBPs) calculated for each window sub-dataset and for all the bipartitions (nodes) reconstructed during the SWB analysis. For example, the SWB_400_ output (SWB analysis based on a window of 400 nt) generated 419,251,473 WBP values, i.e., 780,729 bipartitions with WBP values (between 0% and 100%) for each of the 537 window sub-datasets (Appendix A).

Following [20], the SWB output file was used for SuperTRI analysis [23] to construct the supertree showing the most reliable phylogenetic relationships. Briefly, the LFG program was used to convert the SWB_400_ output into 537 bootstrap log files (lists of bootstrap bipartitions with their WBP values, from 1% to 100%), which were then used as inputs in SuperTRI3 (updated version of SuperTRI [23] for Python3 available at GitHub) to construct an MRP (Matrix Representation with Parsimony) file. The MRP_400_ file is a Nexus matrix of 95 rhinacoviruses and 1,446,681 binary characters (each of them represents a bipartition found during the SWB_400_ analysis); it contains an assumption block to assign the cumulated WBP to all characters. The MRP_400_ file was then executed in PAUP 4.0a [24] using 1000 bootstrap replicates of weighted parsimony (with cumulated WBP values used as weights) to construct a SuperTRI bootstrap 50% majority-rule consensus (SB_400_) tree, which is a supertree showing the phylogenetic relationships supported by the largest genomic fragments. In parallel, we also built SB_500_, SB_600_, SB_1000_, and SB_2000_ supertrees.

### 2.5. Construction of Genomic Bootstrap (GB) Barcodes

A GB barcode is a small image representing the nucleotide alignment in which the genomic regions containing a robust phylogenetic signal (GRPS) were coloured in green, whereas the regions with no (robust) signal were coloured in red. The GB barcodes were constructed for several nodes of interest using the procedure detailed in [20]. The SWB results based on five different window sizes were analysed to identify the intervals of the GRPS, as previously explained in [21]. Then, the CGB program [21] was used to draw the GB barcodes.

### 2.6. Construction of Coloured Genomic Bootstrap (CGB) Barcodes

A phylogenetic CGB barcode is a small image representing the genome of a virus of interest in which the different colours show the best robust phylogenetic signals, i.e., the bipartitions containing the fewest number of closely related viruses, detected in different regions of the alignment. The phylogenetic CGB barcodes were constructed for the five rhinacoviruses found in Vietnam, Ra160660, and the most recent common ancestor [MRCA] of SADS-CoVs using the SWB results based on five window sizes and following the procedure published in [21]. Firstly, the BBC program [20] was used to select only SWB bipartitions showing one or more WBP values ≥ 50% (e.g., 780,729 SWB_400_ bipartitions were reduced to 1252 BBC_400_ bipartitions; Appendix A). Then, only BBC bipartitions including the virus of interest were selected. For example, to reconstruct the CGB of the MRCA of SADS-CoVs, we extracted 293 BBC_400_ bipartitions, 278 BBC_500_ bipartitions, etc. (Appendix A). Then, the bipartitions were ranked in Excel in increasing order of size, from category “+1” (bipartitions including the 34 SADS-CoVs and one closely related virus) to category “+61” (the single bipartition including all viruses of our dataset). To make comparisons between WBPs calculated in the five SWB analyses, all WBP_400_, WBP_500_, WBP_600_, WBP_1000_, and WBP_2000_ ≥ 70% were highlighted in green, and all WBPs between 50% and 70% were highlighted in yellow-green using conditional formatting options in Microsoft^®^ Excel. We performed the comparisons starting with bipartitions of the category “+1”. Due to past events of genomic recombination, we found several bipartitions supporting conflicting phylogenetic relationships, e.g., 34 SADS-CoVs + Ra141388, 34 SADS-CoVs + Rs8462, etc. For each bipartition “+1”, we identified the intervals of genomic regions containing a robust phylogenetic signal (GRPS) using previously published criteria [21]. Then, we proceeded similarly by analysing other genomic fragments for bipartitions “+2” (34 SADS-CoVs + 2 closely related viruses), bipartitions “+3” (34 SADS-CoVs + 3 closely related viruses), etc. In this way, we were able to identify the virus(es) closest to SADS-CoVs in all regions of the genome alignment. In the last step, the intervals of GRPS (5′ and 3′ median positions in the whole-genome alignment) were written in a new CSV file for all bipartitions including the 34 SADS-CoVs in which one or more GRPS were identified as the best phylogenetic signals (i.e., bipartitions containing the fewest number of closely related viruses). A specific colour code was chosen for each bipartition, and the file was used in the CGB program [21] to construct the phylogenetic CGB barcode of the MRCA of SADS-CoVs.

Following [21], the CGB barcodes were used to calculate the phylogenetic contributions of viruses (C_T_), geographic areas (C_TG_), and host species (C_TH_). The exclusive contributions were also calculated for viruses (C_E_), geographic areas (C_EG_), and host species (C_EH_) using only bipartitions showing exclusive ancestry with the virus of interest. For instance, the total geographic contribution (C_TG_) of Yunnan in the CGB barcode of Ra160660 was calculated by summing the GRPS intervals in the whole-genome alignment in which Ra160660 was found to be closely related to one or several viruses from Yunnan and other geographic areas. Then, the sum was multiplied by 100 and divided by the total length of our alignment (27,250 nt) to obtain the percentage contribution. Similarly, the exclusive geographic contribution (C_EG_) of Yunnan was calculated by summing the GRPS intervals in which Ra160660 shared close relationships with one or more viruses from Yunnan only. McNemar’s Chi-squared tests were used to perform pairwise comparisons between the highest geographic contributions, as well as between the highest host contributions.

### 2.7. Recombination and SimPlot Analyses

A recombination analysis was conducted under RDP4.101 [25] to better understand the evolution of several SADS-CoV strains. We ran a full exploratory recombination scan using the seven following methods implemented in RDP4: RDP, GENCONV, BootScan, MaxChi, Chimaera, SIScan, and 3Seq.

Several *Rhinacovirus* genomes were compared with each other using SimPlot++ (https://github.com/Stephane-S/, accessed on 1 December 2023) [26] with a TN93 distance model and a sliding window of 400 nt moving in steps of 50 nt along the whole-genome alignment.

## 3. Results

### 3.1. Rhinacoviruses Detected in Vietnam

Bats were captured in Vietnam during several expeditions in 2021 and 2022 and their faecal samples were previously used for amplifying by RT-qPCR the E gene of *Sarbecovirus*; among the 59 positive samples, 32 were RNA sequenced using the NovaSeq Illumina system to assemble complete *Sarbecovirus* genomes [11]. Five Illumina libraries were found to contain both *Sarbecovirus* and *Rhinacovirus* reads, and it was possible to assemble five complete genomes of *Rhinacovirus* (Table 1). The rhinacoviruses were named using the following rules: the first two letters represent an abbreviation of the bat species; the two first numbers indicate the year of sampling; the two uppercase letters indicate the province of Vietnam (CB: Cao Bang; DB: Dien Bien; QT: Quang Tri) followed by a field number; the “R” letter at the end was used to distinguish *Rhinacovirus* from *Sarbecovirus* names. Based on a BLAST search, we found that the five rhinacoviruses from Vietnam showed between 93% and 98% of nucleotide identity with viruses previously described in China (Appendix A).

### 3.2. Synonymous Nucleotide Composition

The synonymous nucleotide composition (SNC) of the 95 *Rhinacovirus* genomes was analysed at four-fold and two-fold degenerate third-codon positions of all protein-coding genes (alignment length: 26,682 nt). The eight variables (Appendix A) were summarised by a principal component (PC) analysis based on the first two PCs, which contribute 57% and 28% of the total variance, respectively (Figure 2). Five SNC groups can be distinguished: (i) pig SADS-CoVs and related bat rhinacoviruses (*SADSCoVr*) sampled in *R. affinis* and *R. sinicus* from China (Guangdong, Guangxi, Hong Kong, and Yunnan Provinces) and in *R. affinis* and *R. thomasi* from Vietnam (Cao Bang, Dien Bien, and Quang Tri Provinces); (ii) bat rhinacoviruses sampled in *R. affinis* and *R. sinicus* on Hainan Island; (iii) Ra160660 sampled in *R. affinis* from an unspecified province in China; (iv) bat rhinacoviruses sampled in *R. pusillus* from China (Guangdong, Guangxi, Yunnan, and Zhejiang Provinces) and Vietnam (Dien Bien Province); and (v) bat rhinacoviruses sampled in *Myotis laniger*, *Rhinolophus ferrumequinum,* and *Rhinolophus stheno* from Yunnan (here referred to as *YunRhin*). Three of these groups can be diagnosed by specific SNC features: (i) *Rhinacovirus* genomes extracted from *R. pusillus* contain the highest percentages of G and the lowest percentages of U at four-fold degenerate third-codon positions; (ii) *Rhinacovirus* genomes collected on Hainan Island have the highest percentages of C at four-fold degenerate third-codon positions; and (iii) the three *YunRhin* genomes (MlYN15, RfYN2012, and RstYN25) show the highest percentages of U and the lowest percentages of C and G at four-fold degenerate third-codon positions, and the lowest percentages of G at two-fold degenerate third-codon positions.

### 3.3. Phylogenetic Analyses Based on 95 *Rhinacovirus* Genomes

Our alignment of 95 *Rhinacovirus* genomes (27,250 nt) was analysed for phylogeny using two different approaches: (i) a supertree approach was conducted under SuperTRI [23] using SWB results in order to reveal the phylogenetic relationships supported by the largest proportions of the genome [20,21]; and (ii) traditional maximum-likelihood (ML) analyses were carried out under IQ-TREE [19] using whole-genome and Spike gene alignments.

Since the five SB supertrees built using different sizes of sliding windows (400, 500, 600, 1000, and 2000 nt) exhibited high node congruence (Appendix A), we decided to show in Figure 3 their strict consensus tree, in which all topological discordances were collapsed. On each node, we reported the GB barcode in order to better interpret conflicting phylogenetic signals. Our results showed that 67% of the nodes (32 out of 48) were supported by genomic regions representing less than 30% of the genome alignment (GRPS < 30%) and that 29% of the nodes (14 out of 48) were supported by genomic fragments representing more than 40% of the alignment (GRPS ≥ 40%). Among the latter, there are three SNC groups, i.e., rhinacoviruses sampled in *R. pusillus* (GRPS = 92%), rhinacoviruses collected on Hainan Island (GRPS = 89%), and *YunRhin* (GRPS = 79%), as well as the common ancestor of all the 34 SADS-CoVs (GRPS = 45%) and three geographic groups, i.e., rhinacoviruses from *R. pusillus* sampled in Guangxi Province (GRPS = 63%), rhinacoviruses from *R. pusillus* collected in Yunnan Province (southern China) and Dien Bien Province (north-western Vietnam) (GRPS = 48%), and rhinacoviruses from *R. sinicus* collected in Hong Kong (GRPS = 59%).

Overall, the topology of the whole-genome tree inferred with the ML method (Figure 4) was consistent with the SuperTRI consensus. In particular, most nodes supported by GRPS ≥ 40% were found with maximal BP values (100%) in the whole-genome tree. In addition, most nodes of the genome tree showing conflicting relationships with the SuperTRI consensus were not supported by any genomic fragments (GRPS = 0%; Figure 4). The discordant position of RaGD16-Q196 + RaGD17-Q208 as the sister group of RaYN20-Q207 was, however, supported by three regions in ORF1ab (lengths: 500 nt, 600 nt, and 2300 nt; GRPS = 12%; Figure 4). Importantly, this placement makes the bat *SADSCoVr* subgroup from Yunnan (southern China) and Dien Bien (north-western Vietnam) paraphyletic. In the SuperTRI consensus of Figure 3, this *SADSCoVr* subgroup was found to be monophyletic (GRPS = 4%), and RaGD16-Q196 + RaGD17-Q208 appeared to be related to other bat *SADSCoVr* from Guangdong and Guangxi (GRPS = 6%); these relationships were supported by BP = 100% in all the five supertrees (Appendix A).

The Spike gene tree of Figure 5 showed high levels of incongruence with both the SuperTRI consensus and genome trees. The GB barcodes revealed that most of these conflicting relationships were not supported by any genomic fragments (GRPS = 0%) or supported by one or several regions of the Spike gene (GRPS ≤ 8%; positions 20,503–23,941 in GB barcodes). However, there were some exceptions. For instance, the node uniting RaGD19-Q210 and RaGX17-Q212 (BP = 89%) was supported by four genomic regions (GRPS = 13%; Figure 5): two small fragments in ORF1ab (lengths: 350 nt and 250 nt), one fragment of 1400 nt overlapping between ORF1ab and the Spike gene, and another fragment of 1600 nt in the 3′-end of our alignment including both N and NS7a genes. Another example concerned the sister-group relationship between RaYN20-Q207 and RaGD17-Q208, which was supported by four genomic regions (GRPS = 28%; Figure 5): three fragments dispersed in ORF1ab (lengths: 1350 nt, 4000 nt, and 400 nt) and the 5′ half of the Spike gene (length: 1850 nt).

### 3.4. Detection of Several Problematic Genomes

The dichotomy between rhinacoviruses extracted from *R. pusillus* and those sampled in other mammal species (*R. affinis*, *R. sinicus*, *R. thomasi*, *Sus scrofa*, etc.) was found to be supported by 92% of our genome alignment, i.e., regions highlighted in green in the GB barcode of Figure 3. In theory, the two regions coloured in red in positions 14,651–15,400 (ORF1ab gene) and 20,551–21,950 (Spike gene) of the GB barcode could be explained by either a lack of phylogenetic signal (due to high nucleotide conservation among rhinacoviruses) or a discordant phylogenetic signal. By examining SWB bipartitions showing strong support in these two regions, we found that two *Rhinacovirus* genomes sequenced from *R. pusillus* are problematic, RpGX16-Q222 and RpGX17-Q225 (GenBank accessions OQ175208 and OQ175209, respectively [9]): they appeared to be closely related to all SADS-CoVs in positions 14,601–14,900 (SWB_400_ bipartition n°836) and to all SADS-CoVs except CH/FJWT/2018 in positions 21,501–21,900 (SWB_400_ bipartition n°571) (Appendix A); by examining the alignment, we found that they are identical to most SADS-CoVs in positions 14,503–15,555, whereas in positions 21,265–21,959, they are identical to SADS-CoV CHN-GD/2017 and very similar to other SADS-CoVs (99.7% of identity) (Figure 6A). By contrast, both RpGX16-Q222 and RpGX17-Q225 appeared to be very divergent from other members of the *R. pusillus* group in these two regions: between 14% and 15% in the first region and between 27% and 29% in the second region. If we consider RpGX16-Q222 and RpGX17-Q225 genomes to be entirely authentic, we have to admit that a SADS-CoV strain may have recombined with a bat rhinacovirus of the *R. pusillus* group. Such a scenario seems very unlikely as it implies that either pigs or *R. pusillus* bats were coinfected by SADS-CoV and a rhinacovirus of the *R. pusillus* group. As an alternative, we rather suggest that the genomes of RpGX16-Q222 and RpGX17-Q225 are chimeric: although most parts of their genomes are authentic sequences, two regions were mistakenly assembled with SADS-CoV sequences (Figure 6A).

Our analyses revealed that RaYN20-Q207 (GenBank accession: OQ175197 [9]) is another problematic genome. Whereas most parts of its genome were found to support a grouping with RaYN20-Q216 (GRPS = 40%; Figure 3), three fragments of ORF1ab and one fragment of the Spike gene rather provided support for a grouping with RaGD17-Q208 (GRPS = 28%; Figure 5). By examining the alignment, we found a perfect identity between RaYN20-Q207 and RaGD17-Q208 in three genomic regions, i.e., positions 1–1787 (length: 1787 nt), 3820–7943 (length: 4124 nt), and 19,806–20,198 (length: 393 nt), whereas adjacent genomic regions showed much more nucleotide divergence (Appendix A). If these two sequences are authentic, our interpretation is that these three fragments were inherited in the ancestor of RaYN20-Q207 by genomic recombination with a parental virus identical or closely related to RaGD17-Q208. However, this hypothesis seems unlikely, as RaGD17-Q208 was sampled in 2017 in Guangdong, whereas RaYN20-Q207 was collected in 2020 in Yunnan. As an alternative hypothesis, we propose that RaYN20-Q207 is a chimeric genome containing three misassembled fragments from RaGD17-Q208.

The GB barcode reconstructed for the common ancestor of 34 SADS-CoV genomes showed strong phylogenetic support in 45% of our alignment (i.e., regions highlighted in green in the GB barcode of Figure 3). By screening all SWB bipartitions involving SADS-CoVs, we found three SADS-CoV strains showing closer phylogenetic affinities with one or several bat rhinacoviruses in one or two genomic regions (Appendix A): CHN-GD/2017 (Guangdong Province, 2017; GenBank accession: MH539766 [27]), CH/FJWT/2018 (Fujian Province, 2018; GenBank accession: MH615810 [5]), and Guangxi/2021 (Guangxi Province, 2021; GenBank accession: ON911569 [7]).

The 5′-end of CHN-GD/2017 (positions 1–1787) was found to be identical to that of RsHKU2 (GenBank accession: NC_009988) (Figure 6A), whereas these two viruses were collected from a neonatal piglet with acute diarrhoea in 2017 and from a *R. sinicus* in 2006, respectively [12,27]. Since 11 years of evolution without any mutations seems inconceivable, we conclude that the 5′-end of CHN-GD/2017 is not authentic as it was completed with that of the RsHKU2 genome, which was sequenced and published several years before.

In the CH/FJWT/2018 genome, we found two regions supporting a grouping with Ra162140, a bat rhinacovirus, i.e., in positions 20,801–22,250 (Spike gene) and 24,001–24,500 (ORF3) (Appendix A). After alignment inspection, we found that the SADS-CoV CH/FJWT/2018 and Ra162140 genomes share the same sequence in positions 20,634–22,565 (length without gap: 1890 nt) and 23,933–24,638 (length without gap: 703 nt) (Figure 6B). By contrast, they showed less than 96.0% and 97.3% of identity with other SADS-CoVs in the first and second regions, respectively. Although these results may suggest that Ra162140 was a direct parental progenitor of SADS-CoV CH/FJWT/2018, we consider this hypothesis to be unlikely since Ra162140 was collected between 2013 and 2016 from an *R. affinis* of Guangdong Province, China [3,28], whereas SADS-CoV CH/FJWT/2018 was collected in 2018 from a diarrhoeal piglet of Fujian Province, China [5]. After at least two years of evolution and a host species jump from bats to pigs, we indeed expect to find a few mutations separating SADS-CoV CH/FJWT/2018 from Ra162140 in the two problematic regions of 1890 and 703 nt, respectively. As an alternative hypothesis, we, therefore, suggest that SADS-CoV CH/FJWT/2018 could be a chimeric genome: although most parts represent an authentic SADS-CoV sequence, two regions were potentially completed with Ra162140 sequences.

In the genomes of both SADS-CoV CH/FJWT/2018 and Guangxi/2021, we found one small region, in positions 20,251–20,550, supporting a close relationship with four bat rhinacoviruses from Guangdong, i.e., Ra162140, Ra141388, RaGD16-Q195, and RaGD20-Q211 (Appendix A). After alignment inspection, we discovered that all these six viruses share the same sequence in positions 20,017–20,549 (covering the 3′-end of ORF1ab and the first 47 nt of the Spike gene) (Figure 6B). In the same region, other SADS-CoVs show 99.8–100% identity and diverge from bat rhinacoviruses by more than 1.7%. If SADS-CoV sequences CH/FJWT/2018 and Guangxi/2021 are authentic, our interpretation is that both have emerged after recombination between a SADS-CoV strain directly linked to the Guangdong outbreak in 2016–2017 and an undetected SADS-CoV strain closely related to bat rhinacoviruses Ra162140, Ra141388, RaGD16-Q195, and RaGD20-Q211. As an alternative, we propose that the region between positions 20,017 and 20,549 could be non-authentic in both CH/FJWT/2018 and Guangxi/2021 SADS-CoV genomes. This hypothesis is corroborated by the fact that the SADS-CoV CH/FJWT/2018 genome was amplified and sequenced with the Sanger method using 36 sets of unpublished primers [5].

In addition to these potentially non-authentic recombinants, we found many more unique nucleotide signatures in three SADS-CoV genomes, suggesting many sequencing errors. For that, the alignment was analysed in PAUP to find the most parsimonious trees (672 trees of 43,156 steps). The strict consensus tree supported the monophyly of SADS-CoVs and an early divergence of CH/FJWT/2018 followed by that of Guangxi/2021 (Appendix A). The number of autapomorphies was counted for each of the 34 SADS-CoVs. We focused on exclusive autapomorphies (EAs) because they are non-homoplastic (with consistency index = 1) and, therefore, represent unique nucleotide signatures. No more than eight autapomorphies, including four EAs, were found in most SADS-CoV sequences (Appendix A). By contrast, three SADS-CoV genomes exhibited many more autapomorphies and EAs, suggesting multiple sequencing errors: (i) 48 autapomorphies and 26 EAs in the GD-CH/2017 strain (GenBank accession: MG742313 [29]); (ii) 45 autapomorphies and 21 EAs in the Guangxi/2021 strain [7]; and (iii) 15 autapomorphies and 7 EAs in the GD01/2017 strain (GenBank accession: MF370205 [2]). In the CHN-GD/2017 strain (GenBank accession: MH539766 [27]), we found 78 autapomorphies and only a single EA; 57 autapomorphies were located in the 5′-end between positions 238 and 1449 and represent convergences with the bat HKU2 group due to the chimeric nature of MH539766 (see above).

### 3.5. Virus, Geographic, and Host Contributions Calculated for the Ancestor of SADS-CoVs

For this study, the phylogenetic CGB barcode was constructed for the most recent common ancestor (MRCA) of SADS-CoVs using two alignments: the first contains 95 *Rhinacovirus* genomes (including 34 SADS-CoVs) and the second contains only 87 *Rhinacovirus* genomes (including 29 SADS-CoVs) as the eight problematic genomes identified in Section 3.4 were excluded.

The contributions of bat rhinacoviruses in the phylogenetic CGB barcode of the MRCA of 34 SADS-CoVs (calculated on the dataset of 95 genomes) are shown in Figure 7A. The highest contributions (C_T_ > 50%) were found for rhinacoviruses from Guangdong sampled in *R. affinis* and *R. sinicus*, including Ra141388 (C_T_ = 81%; C_E_ = 5%), Ra162140 (C_T_ = 78%; C_E_ = 1.8%), RaGD20-Q211 (C_T_ = 68%; C_E_ = 0.7%), Rs8495 (C_T_ = 67%), RaGD16-Q195 (C_T_ = 63%), Rs8462 (C_T_ = 61%; C_E_ = 3.9%), and RaGD17-Q209 (C_T_ = 54%; C_E_ = 0.6%). For all other viruses, the contributions were less important (≤41%), and we did not detect any genomic fragment supporting exclusive ancestry. The contribution of bat rhinacoviruses from Guangdong was found to be significantly higher (at 0.01 level) than those from other Chinese provinces (Figure 7B): C_TG_ = 99% with C_EG_ = 51% versus C_TG_ ≤ 42% with C_EG_ = 1% for Guangxi and 0% for all other provinces. The two bat species showing the highest phylogenetic contributions for SADS-CoV were *R. affinis* (C_TH_ = 96%; C_EH_ = 23%) and *R. sinicus* (C_TH_ = 74%; C_EH_ = 4%). The contributions of other bat species were much less important (C_TH_ ≤ 35%; C_EH_ ≤ 1%) (Figure 7C). When the eight problematic genomes presented previously were excluded from SWB and CGB analyses, the contribution of *R. affinis* was still found to be significantly more important than that of other species (C_TH_ = 94% versus C_TH_ ≤ 64%; C_EH_ = 35% versus C_EH_ ≤ 5%) (Table 2). Similarly, the contribution of viruses from Guangdong was still found to be significantly more important (C_TG_ = 99%; C_EG_ = 65%) than those of all other geographic areas (C_TG_ ≤ 30%; C_EG_ ≤ 1%) (Table 3). Therefore, we conclude that the ancestor of SADS-CoVs emerged in *R. affinis* from Guangdong.

### 3.6. On the Geographic Sampling of Ra160660

Although the genome of Ra160660 was published, the authors did not provide any information on its province of origin in China [8]. This is problematic as our analyses revealed that Ra160660 has some specific genomic features: (i) based on both SuperTRI and genome trees, it belongs to the *SADSCoVr* group (BP_SB_ = 100% in all the five supertrees, Appendix A; GRPS = 20%, Figure 3; BP = 100%, Figure 4); (ii) based on the Spike gene tree, Ra160660 was found to be the sister group of *YunRhin* (BP = 99%); the GB barcode showed that this relationship was supported by 6% of our alignment (Figure 5); (iii) the SNC was found to be different from that of the *SADSCoVr* and *YunRhin* groups (Figure 2). The phylogenetic CGB barcode of Ra160660 revealed that viruses of the *SADSCoVr* group provide many more contributions than those of the *YunRhin* group (C_T_ = 93% versus 58%; C_E_ = 37% versu*s* 7%). These results, therefore, indicate that Ra160660 appeared by recombination between *SADSCoVr* and *YunRhin* parental viruses. In addition, the contribution of viruses from Yunnan was found to be significantly more important than those of other geographic areas (C_T_ = 96% versus ≤70%; C_E_ = 28% versus ≤1%; Table 3). This result agrees with its placement within a *SADSCoVr* subgroup of Yunnan (BP_SB_ = 100%; GRPS = 3%, Figure 3). We conclude that Ra160660 was collected in Yunnan or in an ecologically related province not sampled in our study, such as Sichuan or Guizhou. Our prediction was confirmed by one of our reviewers, who informed us that Ra160660 was collected in Yunnan in 2016.

### 3.7. Phylogeography of Bat Rhinacoviruses from Vietnam

In our study, we assembled five complete *Rhinacovirus* genomes from three different localities in Vietnam. Although our sampling efforts for rhinacoviruses need to be increased in Vietnam, this is a good opportunity to test geographic and host affinities with viruses previously collected in China. For that purpose, the CGB barcodes were reconstructed for all the five *Rhinacovirus* genomes detected in Vietnam, and their virus, geographic, and host contributions were calculated (Table 2 and Table 3). All three rhinacoviruses sampled in Dien Bien Province in north-western Vietnam (Ra22DB107R, Ra22DB163R, and Rp22DB167R) were found to share more affinities with rhinacoviruses previously collected in Yunnan, China: between 88% and 94% of their best phylogenetic signals involved viruses from Yunnan, and between 21% and 48% of their genomes showed exclusive relationships with viruses from Yunnan. These results are congruent with the SuperTRI consensus of Figure 3, in which Ra22DB107R, Ra22DB163R, and Rp22DB167R were found to be closely related to viruses from Yunnan. Although it was found to be paraphyletic in the genome tree of Figure 4, the bat *SADSCoVr* subgroup from Yunnan (southern China) and Dien Bien (north-western Vietnam) was recovered monophyletic in both the genome tree and SuperTRI consensus tree based on 87 *Rhinacovirus* genomes, i.e., excluding the eight problematic sequences (Appendix A, respectively).

Similarly, Rt22CB395R, which was sampled in Cao Bang (north-eastern Vietnam), showed more affinities with viruses previously collected in Guangxi and Guangdong Provinces (southern China), which fully agrees with their placements in phylogenetic trees based on 95 and 87 *Rhinacovirus* genomes (Figure 3 and Figure 4; Appendix A).

The phylogeographic affinities of Rt22QT46R, which was collected in Quang Tri (central Vietnam), were more difficult to decipher. Indeed, several geographic areas (north-eastern Vietnam, north-western Vietnam, Yunnan, Guangxi, and Guangdong), as shown in Table 3, had high values of total contribution (87–91%) and low values of exclusive contribution (0–2%), suggesting that Rt22QT46R belongs to a lineage distinct from the two other geographic groups of *SADSCoVr*, i.e., (i) Dien Bien and Yunnan and (ii) Cao Bang, Guangxi, and Guangdong.

## 4. Discussion

### 4.1. Host and Geographic Evolution of the Four Divergent *Rhinacovirus* Groups

Our phylogenetic and SNC analyses provided strong support to recognize four groups of *Rhinacovirus*: (i) viruses sampled in *R. pusillus* from China (Guangdong, Guangxi, Yunnan, and Zhejiang Provinces) and north-western Vietnam (Dien Bien Province); (ii) viruses sampled in *R. affinis* and *R. sinicus* collected on Hainan Island; (iii) viruses sampled in several bat species (*M. laniger*, *R. ferrumequinum*, and *R. stheno*) from Yunnan, a group hereafter named *YunRhin*; and (iv) *SADSCoVr*, a large group including SADS-CoV strains extracted from domestic pigs from several provinces in China (Guangdong, Fujian, Jiangxi, and Guangxi) and rhinacoviruses sampled in three *Rhinolophus* species (*R. affinis*, *R. sinicus*, and *R. thomasi*) collected in several provinces of southern China (Guangdong, Guangxi, and Yunnan) and Vietnam (Cao Bang, Dien Bien, and Quang Tri). The *Rhinacovirus* genomes from Hainan and *SADSCoVr* groups showed more similar SNCs than those extracted from *R. pusillus* and *YunRhin* groups (Figure 2). Despite that, our phylogenetic analyses rather support a sister-group relationship between *SADSCoVr* and *YunRhin* (GRPS = 27%, Figure 3; BP = 98%, Figure 4). These results suggest, therefore, that the *YunRhin* group has evolved divergently in Yunnan after a jump from a bat species assemblage composed of *R. affinis*, *R. sinicus,* and *R. thomasi* to another bat species assemblage containing *R. stheno*, *R. ferrumequinum,* and *M. laniger*. It is interesting to note that a similar situation was previously described for the subgenus *Sarbecovirus*, in which a highly divergent group endemic to Yunnan, called *YunSar*, was described for viruses mainly sampled in *R. stheno* [17,20]. We proposed, therefore, the following scenario for the evolution of rhinacoviruses in bats: a deep host separation between *R. pusillus* and a species assemblage composed of *R. affinis*, *R. sinicus*, and *R. thomasi*; in the latter group, an allopatric evolution of rhinacoviruses on Hainan Island due to the isolation of their hosts by the Qiongzhou Strait (30 km) and, in parallel, a jump to a new species assemblage (including *R. stheno* and *R. ferrumequinum nippon*) in Yunnan. In the *SADSCoVr* group, our analyses also suggested a west/east division separating the rhinacoviruses of Yunnan (south-western China) and Dien Bien (north-western Vietnam) from those of Cao Bang (north-eastern Vietnam), Guangxi, and Guangdong (south-eastern China). This division fits well with two ecoregions ([30], https://ecoregions.appspot.com, accessed on 1 May 2024): in the west, the northern Indochina subtropical forests (Eco ID: 256), which includes the highlands of southern Yunnan, north-western Vietnam, north-eastern Myanmar, and northern Laos; in the east, the South China–Vietnam subtropical evergreen forests (Eco ID: 268), which extends from northern Vietnam into south-eastern China. It could be explained by the high mountainous range (>2000 m) between the Black and Red Rivers in Yunnan (Ailao Shan mountains) and north-western Vietnam (Hoang Lien Son), as these biogeographic barriers can limit the dispersal capacity of some bat populations.

### 4.2. SADS-CoV Evolved from *Rhinolophus affinis* Circulating in Guangdong

Our phylogeographic analyses of *Rhinacovirus* genomes provided strong evidence that the MRCA of SADS-CoV strains originated in *R. affinis* living in Guangdong Province: (i) the phylogeographic contribution of Guangdong was significantly higher than that of Vietnam and other provinces in China (Table 3); (ii) all viruses showing the highest phylogenetic contributions came from bats captured in Guangdong (Figure 7); and (iii) the host species showing the best phylogenetic contributions is *R. affinis* (C_TH_ = 94–96%; C_EH_ = 23–35%), followed by *R. sinicus* (C_TH_ = 64–74%; C_EH_ = 4–5%). An emergence of SADS-CoV in bats circulating in Guangdong is perfectly consistent with the first documented outbreak in pigs, which occurred between August 2016 and May 2017 in several pig farms, all located in northern Guangdong [3,4]. Like other enteric viruses, rhinacoviruses are transmitted through the faecal–oral route [10], which means that cross-species infection can occur after contact with the diarrhoea or vomitus of contaminated host species. Therefore, the first bat-to-pig transmission may have involved food contaminated by bat guano or the consumption of bat corpses, either when bats visited pig farms or when pigs visited bat caves. Free-ranging pig farming is a long-standing practice in many areas across Asian countries, including southern China and Vietnam. During the winter, such pigs generally choose to avoid the cold by occupying caves, where horseshoe bats are likely to hibernate. Thus, SADS-CoV may have circulated among free-ranging pigs in some remote villages well before the first case was detected in modern farms in 2016.

### 4.3. On the Re-Emergences of SADS-CoV in Fujian in 2018 and Guangxi in 2021

As for other coronaviruses, the large RNA genomes of rhinacoviruses evolve through two main processes, nucleotide mutation and inter-genomic recombination, but their impacts on the evolution of bat and swine rhinacoviruses remain poorly studied. Using the RDP4 package, researchers [7] identified in the SADS-CoV Guangxi/2021 genome one putative recombination event involving two SADS-CoV lineages and a breakpoint at the junction between ORF1ab and the Spike gene. However, only four bat rhinacoviruses were included in their study. By analysing our alignment of 95 *Rhinacovirus* genomes under RDP4, we estimated 286 recombination events and 2095 recombination signals. Among them, several recombination signals involving both swine and bat rhinacoviruses were detected in three SADS-CoV strains, i.e., CHN-GD/2017, CH/FJWT/2018, and Guangxi/2021. Our analyses based on GB barcodes and pairwise distances (Appendix A) revealed that all these recombination signals are due to SADS-CoV genomic fragments that are 100% identical to previously published bat rhinacoviruses; therefore, we question their authenticity: SADS-CoV CHN-GD/2017 was identical to RsHKU2 [12] in positions 1–1787; SADS-CoV CH/FJWT/2018 was identical to Ra162140 [3] in positions 20,634–22,565 and 23,933–24,638; SADS-CoV Guangxi/2021 was identical to SADS-CoV CH/FJWT/2018 [5] and four bat rhinacoviruses, including Ra162140 [3], in positions 20,017–20,549 (covering the 3′-end of ORF1ab and the 5′-end of the Spike gene).

As explained in Section 3.4, there is no doubt that the SADS-CoV CHN-GD/2017 genome was misassembled with the 5′-end of RsHKU2 as the latter genome was probably used as a reference for constructing the consensus sequence. For the two other SADS-CoV strains, two opposing hypotheses, i.e., natural recombination versus human errors, can be formulated because the bat rhinacoviruses showing a perfect identity in one or several genomic fragments were those identified as the most closely related to the MRCA of SADS-CoVs, i.e., Ra162140 and Ra141388 (Figure 3 and Figure 4). According to the hypothesis involving recombination, CH/FJWT/2018 and Guangxi/2021 genomes are considered to be fully authentic; based on the three genomic fragments A, B and B’ shown in Appendix A, we can propose that at least three recombination events occurred in pigs and involved the main SADS-CoV lineage (the one containing all strains currently sequenced except CH/FJWT/2018 and Guangxi/2021) and a second swine lineage identical or more closely related to the bat rhinacovirus Ra162140. According to the hypothesis involving human errors, CH/FJWT/2018 and Guangxi/2021 genomes are considered to be artificially chimeric; the fragment(s) of bat rhinacoviruses found in their genomes were not authentic; they were integrated into the sequences following human errors, which could be due to carry-over contamination during PCRs, incorrect pooling of pig and bat samples in the same NGS library, or bioinformatic error (e.g., the consensus sequence was misassembled by calling a bat reference sequence when there was no coverage). To provide definitive conclusions on the authenticity of their genomes, we consider that [5,7] should deposit their metagenomic data and Sanger electropherograms in international databases and publish the primer sets they used for PCR amplification and sequencing. In the absence of these data, these two problematic SADS-CoV genomes should be considered doubtful and excluded from future studies.

When the five problematic SADS-CoV genomes (highlighted with black and grey circles in Figure 3) were excluded from the analyses, it appeared that the 29 remaining SADS-CoVs share more than 99.87% of genomic identity. Importantly, they included 23 SADS-CoVs collected by different research teams in Guangdong during the 2016–2017 outbreak [1,3,14,31,32], as well as two SADS-CoVs sampled in Guangdong in 2018 (unpublished), three SADS-CoVs sampled in Jiangxi in 2018 (unpublished), and one SADS-CoV sampled in Guangdong in February 2019 [6]. Recently, [33] reported that SADS-CoV re-emerged in June 2023 in Henan Province (central China) with an outbreak that caused the death of about 400 piglets. The genome of the new SADS-CoV strain (HNNY/2023; GenBank accession: PP069800 [33]) shared 99.92% of nucleotide identity with the first SADS-CoVs identified in Guangdong in 2017 [3]. Such a high nucleotide conservation means that all SADS-CoV strains originated from the same source, i.e., farmed pigs contaminated by bats in northern Guangdong, and that the virus has been circulating silently in domestic pigs for at least seven years, spreading from farm to farm in Guangdong between 2016 and 2019 [1,2,3,4], and then in the west in Guangxi in 2021 [7], and in the north in Jiangxi and Fujian in 2018 [5,10], and more recently in Henan in 2023 [33]. To stop the spread, we recommend a multi-day quarantine followed by diagnostic testing for SADS-CoV before the transport of live pigs.

## Figures and Tables

**Figure 1 viruses-16-01114-f001:**
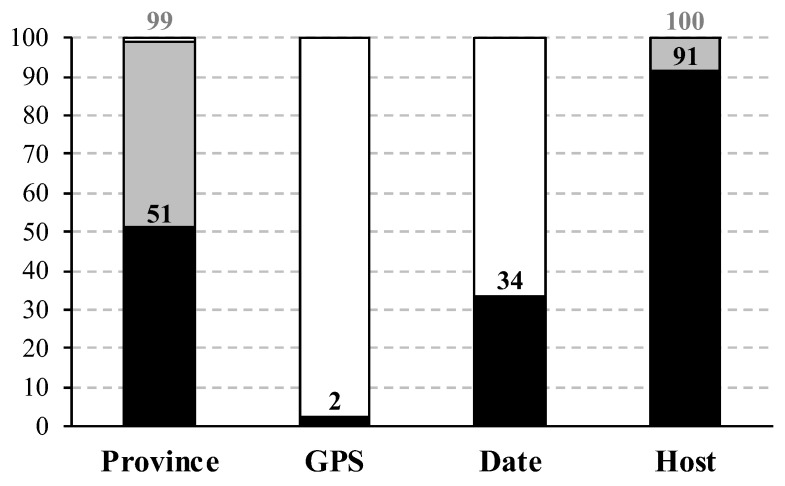
Percentages of available data for the 92 *Rhinacovirus* genomes previously sequenced in China. Data of interest: geographic origin (province name and Global Positioning System [GPS] coordinates), date of field sampling (day, month, year), and binomial name of the host species. The percentages of data available in GenBank are shown by black histograms. Some additional data were extracted from the scientific literature, and the cumulated percentages are indicated by grey histograms.

**Figure 2 viruses-16-01114-f002:**
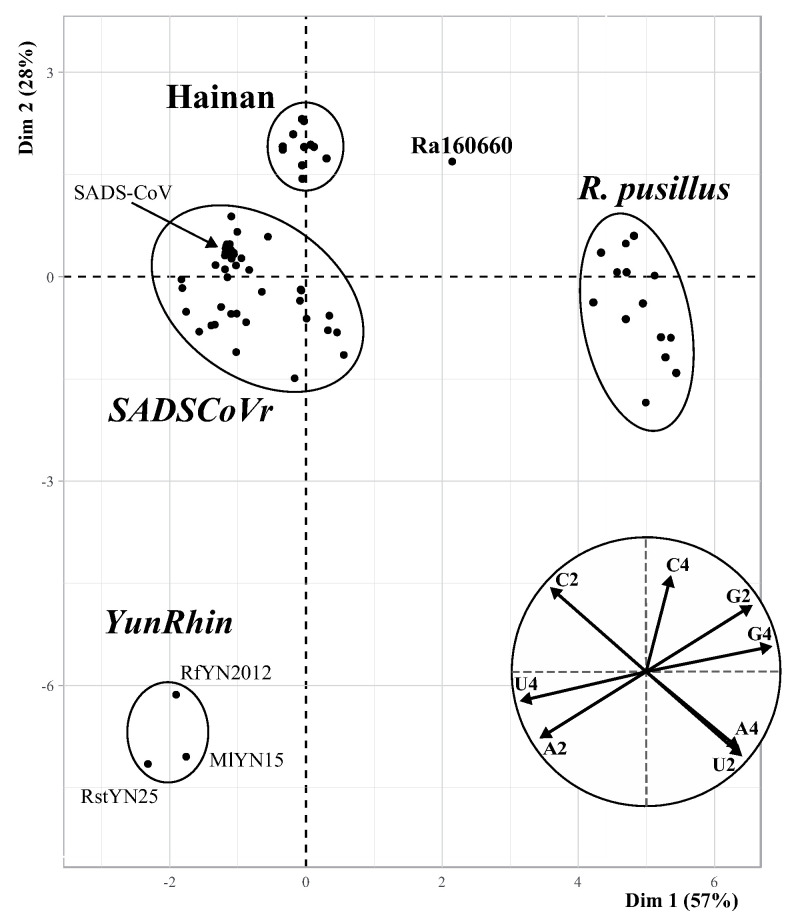
Variation in synonymous nucleotide composition (SNC) at third-codon positions of *Rhinacovirus* genomes. The alignment of protein-coding genes (26,682 nt; 95 *Rhinacovirus* genomes) was used to calculate the frequency of the four bases at either four-fold degenerate third-codon positions (A4, C4, G4, and U4 variables) or two-fold degenerate third-codon positions for either purines (A2 and G2 variables) or pyrimidines (C2 and U2 variables) (Appendix A). The main graph represents the individual factor map obtained from the principal component (PC) analysis based on the eight variables. The small circular graph at the bottom right represents the variables factor map. The five SNC groups here identified are (i) swine and bat viruses related to SADS-CoV (here referred to as *SADSCoVr*), (ii) bat rhinacoviruses from Hainan (Hainan group), (iii) the bat rhinacovirus Ra160660 (GenBank accession: MN611522 [8]), (iv) bat rhinacoviruses sampled in *Rhinolphus pusillus* (*R. pusillus* group), and (v) bat rhinacoviruses from Yunnan Province showing a divergent SNC (*YunRhin*). Four *Rhinacovirus* genomes collected in Vietnam belong to *SADSCoVr* (Ra22DB107R, Ra22DB163R, Rt22CB395R, and Rt22QT46R), while RpDB167R belongs to the *R. pusillus* group.

**Figure 3 viruses-16-01114-f003:**
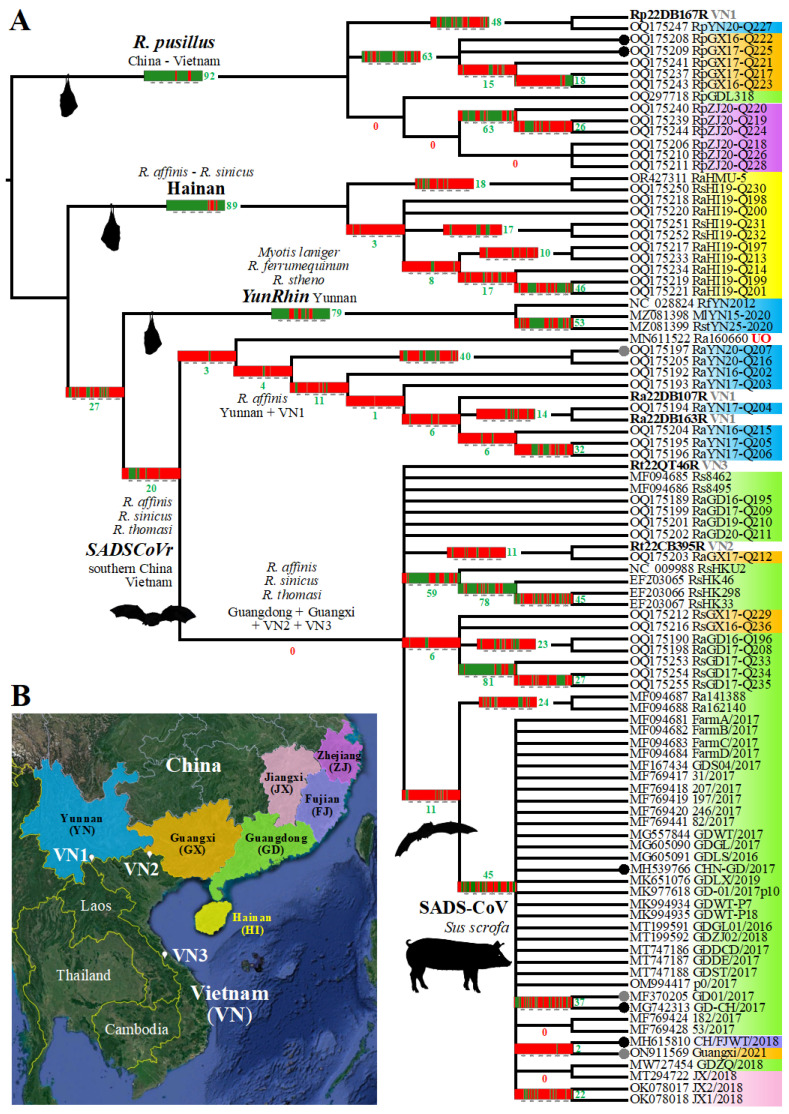
SuperTRI consensus tree built from five SWB analyses based on an alignment of 95 *Rhinacovirus* genomes. The alignment of 27,250 nucleotides was analysed using the SWB program [20] and five different window sizes (400, 500, 600, 1000, and 2000 nt). Then, the five SWB output files were transformed into five MRP files (with LFG and SuperTRI programs [20,23]), which were then executed in PAUP 4.0 [24] to construct five SuperTRI bootstrap 50% majority-rule consensus (SB) supertrees using weighted parsimony and 1000 bootstrap replicates. The tree shown (**A**) is a strict consensus of the five SB supertrees based on different window sizes. The five bat rhinacoviruses from Vietnam are written in bold. The background colours indicate the Chinese provinces where the viruses were sampled (see map in (**B**)). The eight *Rhinacovirus* genomes highlighted by a black or grey circle at the left of their accession number were found to contain non-authentic genomic regions (black circles) or were found to be potentially problematic (grey circles) (see text for more details). The genomic bootstrap (GB) barcodes were constructed by identifying the intervals of genomic regions containing a robust phylogenetic signal (GRPS, shown in green in GB barcodes; regions in red do not support the phylogenetic signal). The intervals of the GRPS were used to calculate the proportion of the genome alignment supporting phylogenetic relationships (value in green for each GB barcode). Internal branches marked by a red zero were not supported by any genomic fragment (GRPS = 0%). The map (**B**) shows the Chinese provinces and the three geographic localities in Vietnam (VN1, VN2, and VN3). Map from Google Earth Pro (version 7.3.3.7786), US Dept of State Geographer © 2020 Google—Image Landsat/Copernicus—Data SIO, NOAA, US Navy, NGA, GEBCO.

**Figure 4 viruses-16-01114-f004:**
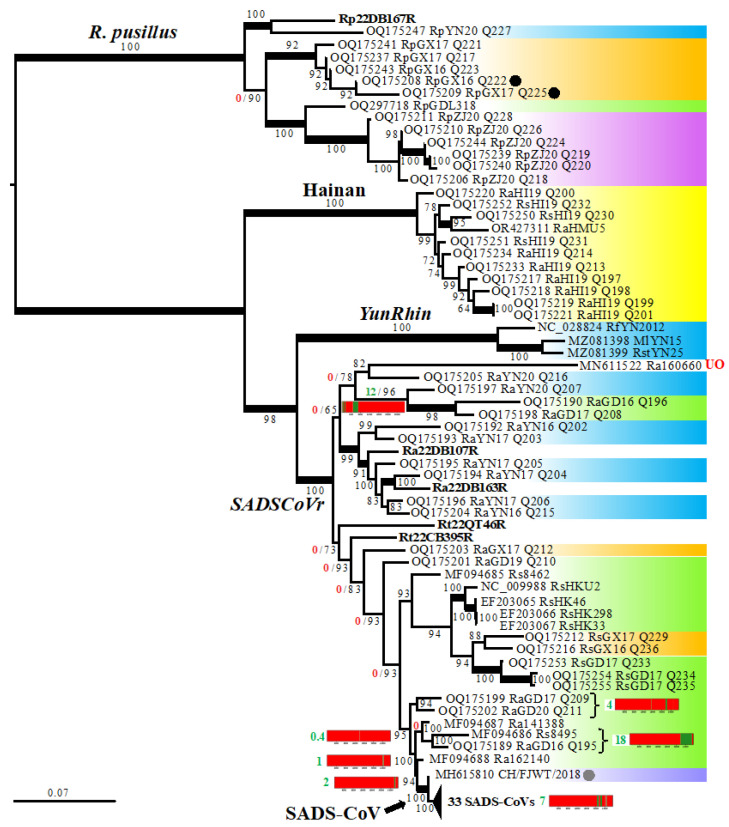
Maximum-likelihood tree of 95 *Rhinacovirus* genomes. The alignment of 27,250 nucleotides was analysed using IQtree [19] and 1000 bootstrap replicates. Nodes consistent with the supertree consensus of Figure 3 are indicated by thicker branches. The GB barcodes are shown for a selection of nodes in conflict with the supertree consensus, i.e., those including SADS-CoVs and those grouping rhinacoviruses from different Chinese provinces. See legend of Figure 3 for more details.

**Figure 5 viruses-16-01114-f005:**
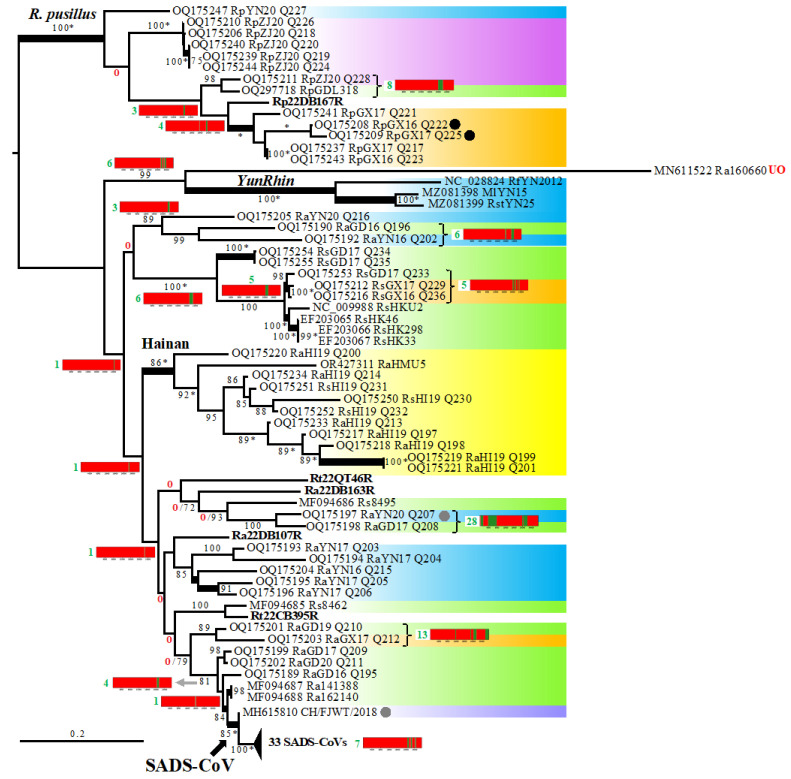
Maximum-likelihood tree of 95 *Rhinacovirus* Spike genes. The alignment of 3435 nucleotides was analysed using IQtree [19] and 1000 bootstrap replicates. Nodes consistent with the supertree consensus of Figure 3 are indicated by thicker branches; those congruent with the whole-genome tree of Figure 4 are indicated by asterisks. The GB barcodes are shown for a selection of nodes in conflict with the supertree consensus, i.e., those including SADS-CoVs and those grouping rhinacoviruses from different Chinese provinces. See legend of Figure 3 for more details.

**Figure 6 viruses-16-01114-f006:**
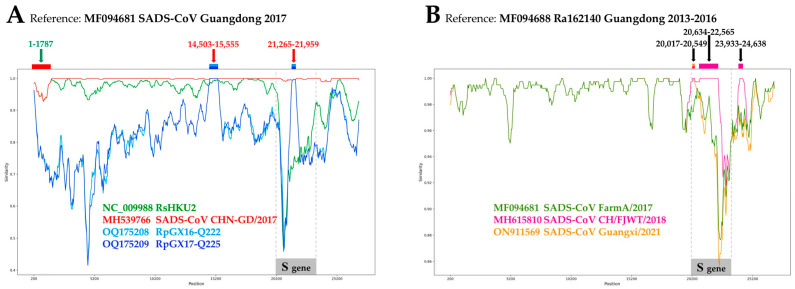
Similarity plot analyses of five problematic *Rhinacovirus* genomes (GenBank accession numbers written in bold below). The problematic *Rhinacovirus* genomes were detected following the study of SWB (sliding window bootstrap) bipartitions (see text for more details). The similarity plot analyses were performed with the SimPlot++ program [26]. In (**A**), one of the SADS-CoV genomes sequenced in [3] (GenBank accession: MF094681; Guangdong Province, 2017) was used as a reference for pairwise comparisons with RsHKU2 (NC_009988), one problematic SADS-CoV genome (Guangdong Province, 2017; **MH539766** [27]), and two problematic *Rhinacovirus* genomes sequenced from *Rhinolophus pusillus* collected in Guangxi Province, China (**OQ175208** and **OQ175209** [9]). In (**B**), the Ra162140 genome sequenced from *Rhinolophus affinis* collected in Guangdong (MF094688 [3]) was used as reference for pairwise comparisons with three SADS-CoV genomes, including MF094681 (see above) and two potentially problematic sequences, **MH615810** (Fujian Province, 2018 [5]) and **ON911569** (Guangxi Province, 2021 [7]).

**Figure 7 viruses-16-01114-f007:**
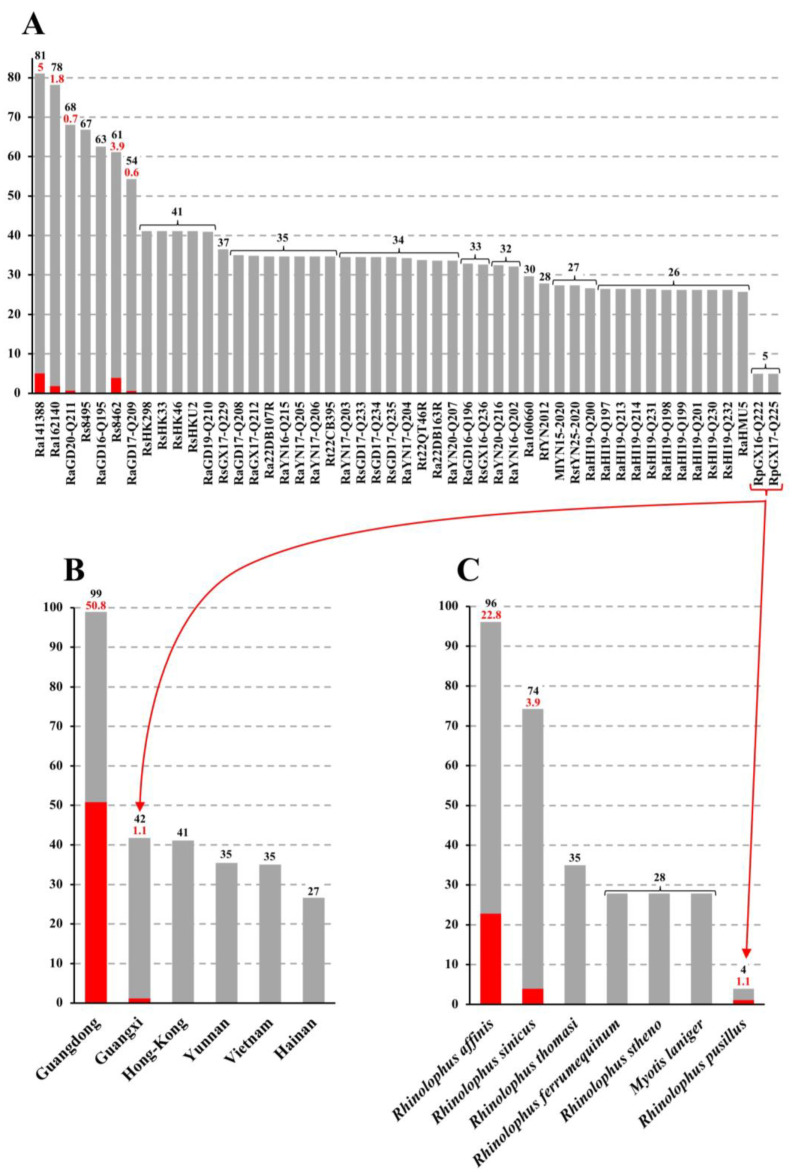
Virus (**A**), geographic (**B**), and host species (**C**) contributions calculated for the most recent common ancestor (MRCA) of SADS-CoV. The coloured genomic bootstrap (CGB) barcode was reconstructed for the most recent common ancestor (MRCA) of 34 SADS-CoVs using an alignment of 95 *Rhinacovirus* genomes and the method published in [21]. Then, we calculated the total contribution (C_T_) and exclusive contribution (C_E_) of bat rhinacoviruses (**A**), geographic areas corresponding to Chinese provinces and Vietnam (**B**), and bat host species (**C**). The total contributions correspond to the percentages of the genome alignment for which the best phylogenetic signals involved the virus, geographic area, or host species of interest (grey histograms, values written in black). The exclusive contributions correspond to the percentages of the genome alignment for which the virus, geographic area, or host species of interest was the only one involved in the best phylogenetic signals (red histograms, values written in red).

**Table 1 viruses-16-01114-t001:** *Rhinacovirus* genomes assembled in this study. (Abbreviation: Loc. = locality; *R.* = *Rhinolophus*; S = sex; VN = Vietnam).

Virus Name	Date	Sample Code(s)	Host Species	S	Loc.	NovSeq Reads	Virus Reads	Length (nt)	GenBank
Ra22DB107R	24 June 2022	DB107	*R. affinis*	♀	VN1	147,368,172	76,009	27,137	PP746501
Ra22DB163R	26 June 2022	DB163	*R. affinis*	♀	VN1	221,895,122	6084	27,130	PP746502
Rp22DB167R	26 June 2022	DB167	*R. pusillus*	♀	VN1	167,424,614	51,354	27,149	PP746503
Rt22CB395R	28 February 2022	PO395+399	*R. thomasi*	♀	VN2	152,809,588	3,159,553	27,165	PP746500
Rt22QT46R	2 November 2022	QT46+47+55+57	*R. thomasi*	♂	VN3	192,686,690	17,037,975	27,129	PP746504

**Table 2 viruses-16-01114-t002:** Total and exclusive host contributions (in percentages; at the left and right of the slash, respectively) calculated on coloured genomic bootstrap (CGB) barcodes reconstructed for a selection of viruses.

	*Myotis* *laniger* (*n* = 1)	*R.* *affinis* (*n* = 28)	*R. f. nippon ** (*n* = 1)	*R.* *pusillus* (*n* = 14)	*R.* *sinicus* (*n* = 14)	*R.* *stheno* (*n* = 1)	*R.* *thomasi* (*n* = 2)	*Sus* *scrofa* (*n* = 34)
SADS-CoV ^1^	28/0	**96**/**23**	28/0	4/1	74/4	28/0	35/0	NA
SADS-CoV ^2^	16/0	**94**/**35**	17/0	0/0	64/5	16/0	21/1	NA
Ra22DB107R	28/0	**98**/**30**	39/2	1/0	37/0	28/0	50/0	37/0
Ra22DB163R	22/0	**100**/**57**	30/0	1/0	26/0	22/0	36/0	25/0
Rp22DB167R	33/0	33/0	33/0	**100**/**67**	33/0	33/0	33/0	33/0
Rt22CB395R	30/0	**90**/**38**	31/0	1/0	59/7	30/0	46/2	49/0
Rt22QT46R	55/0	**95**/3	59/0	6/0	88/2	55/0	87/2	83/0
Ra160660	55/0	**91**/**25**	63/0	2/0	61/0	55/0	65/1	58/0

^1^: Common ancestor of 34 swine rhinacoviruses; ^2^: common ancestor of 29 swine rhinacoviruses (dataset reduced to 87 genomes; see main text for details); *: *R. ferrumequinum nippon*; bold values: statistically significant at 0.01 level; abbreviations: *R*: *Rhinolophus*.

**Table 3 viruses-16-01114-t003:** Total and exclusive geographic contributions (in percentages; at the left and right of the slash, respectively) calculated on coloured genomic bootstrap (CGB) barcodes reconstructed for a selection of viruses.

	NEV (*n* = 1)	CV (*n* = 1)	NWV (*n* = 3)	VN (*n* = 5)	YN (*n* = 12)	GX (*n* = 8)	GD (*n* = 14)	HK (*n* = 4)	HI (*n* = 11)	ZJ (*n* = 6)
SADS-CoV ^1^	35/0	34/0	34/0	35/0	35/0	42/1	**99**/**51**	41/0	27/0	0/0
SADS-CoV ^2^	21/1	20/0	21/1	22/1	20/0	24/0	**99**/**65**	30/0	16/0	0/0
Ra22DB107R	49/0	35/0	74/6	74/6	**94**/**21**	37/0	37/0	33/0	25/0	0/0
Ra22DB163R	34/0	27/0	56/6	61/6	**89**/**34**	31/4	26/0	26/0	19/1	1/0
Rp22DB167R	33/0	33/0	33/0	33/0	**88**/**48**	50/6	39/0	33/0	33/0	8/0
Rt22CB395R	NA	46/2	62/0	65/2	60/3	68/11	70/14	47/0	26/0	0/0
Rt22QT46R	87/2	NA	87/0	90/2	91/0	89/0	90/2	84/0	45/0	2/0
Ra160660	61/0	64/1	69/0	70/1	**96**/**28**	62/0	61/0	60/1	39/0	2/0

^1^: Common ancestor of 34 swine rhinacoviruses; ^2^: common ancestor of 29 swine rhinacoviruses (dataset reduced to 87 genomes; see main text for details); bold values: statistically significant at 0.01 level; abbreviations: CV: central Vietnam; GD: Guangdong Province; GX: Guangxi Province; HK: Hong Kong; HI: Hainan Island; NEV: north-eastern Vietnam; NWV: north-western Vietnam; ZJ: Zhejiang Province; YN: Yunnan Province.

## Data Availability

The meta-transcriptomic sequencing reads generated in this study have been deposited in the SRA database under accession code PRJNA1025946. The viral genome sequences have been deposited in GenBank with the accession numbers PP746500–PP746504. The datasets generated and analysed during the current study, including the whole-genome alignment, SWB files, BBC files, SuperTRI files (log files used as inputs for SuperTRI, MRP files), and CGB files are available in the Open Science Framework (OSF) platform at https://osf.io/n3svc (created on 03 July 2024). The SWB, BBC, CGB, and LFG programs are available at https://github.com/OpaleRambaud/GB_barcodes_project (created on 12 May 2023). The SuperTRI3 program is available at https://github.com/thaninacbn/SuperTRI3 (created on 30 April 2024).

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
