# Peer review of "Bat Rhinacoviruses Related to Swine Acute Diarrhoea Syndrome Coronavirus Evolve under Strong Host and Geographic Constraints in China and Vietnam"

_viruses, 2024, doi:10.3390/v16071114_

Round 1

Reviewer 1 Report

Comments and Suggestions for Authors

The presented manuscript describes the detailed phylogenetic analysis of the recently discovered swine virus and its relation to the bat viruses from which it most likely originated. This study adds new data to the understanding of the coronavirus family evolution.

Author Response

Thanks for the review.

Reviewer 2 Report

Comments and Suggestions for Authors

Dear authors, 

The proposed work is an interesting analysis of the SADS-CoV virus using original data (five original sequences from bats) and external data from GenBank (95 additional sequences). The manuscript describes the analysis of the synonymous nucleotide composition, the phylogenetic analysis and the analysis of potential recombinants. The research problem is current and well-presented. The methods are well-documented, and the conclusion is sound. Also, the putative reconstruction of the events originating the contamination between bats and pigs is particularly interesting. However, I found the reading difficult due to the abundance of acronyms and details in the text. In case the other reviewers request a major revision, I suggest moving some details of the analysis as supplementary material and lightening the main text. While the manuscript can be improved, the work is worthy of publication. 

I suggest the authors to correct the points reported below: 

========================== BEGIN OF COMMENTS =====================

In the Introduction, lines 60/61, the authors mention the taxonomy of SADS-CoV without precisely referencing the taxon entry in the NCBI Taxonomy database or ICTV. I suggest adding the taxon ID corresponding to this virus in NCBI and ICTV. Taxa IDs may be helpful for the reader because a simple search of the virus name in both systems does not return the correct taxon. Indeed, the sequences used in this study and their corresponding taxa were found through a BLAST search, as later referenced in line 99. 

In the Discussion section 4.2 (SADS-CoV evolved from Rhinolophus affinis circulating in Guangdong), line 645, the authors write: "Our phylogeographic analyses based on either 95 or 87 Rhinacovirus genomes [...]". I understand that eight problematic sequences have been removed from the initial dataset, as described in Section 3.7. However, I suggest either removing "or 87" or repeating in the Discussion the reason that excluded those eight sequences. 

========================== END OF COMMENTS =====================

Kind regards.

Author Response

Dear authors,

The proposed work is an interesting analysis of the SADS-CoV virus using original data (five original sequences from bats) and external data from GenBank (95 additional sequences). The manuscript describes the analysis of the synonymous nucleotide composition, the phylogenetic analysis and the analysis of potential recombinants. The research problem is current and well-presented. The methods are well-documented, and the conclusion is sound. Also, the putative reconstruction of the events originating the contamination between bats and pigs is particularly interesting. However, I found the reading difficult due to the abundance of acronyms and details in the text. In case the other reviewers request a major revision, I suggest moving some details of the analysis as supplementary material and lightening the main text. While the manuscript can be improved, the work is worthy of publication.

Thanks for the comments.

I suggest the authors to correct the points reported below:

In the Introduction, lines 60/61, the authors mention the taxonomy of SADS-CoV without precisely referencing the taxon entry in the NCBI Taxonomy database or ICTV. I suggest adding the taxon ID corresponding to this virus in NCBI and ICTV. Taxa IDs may be helpful for the reader because a simple search of the virus name in both systems does not return the correct taxon. Indeed, the sequences used in this study and their corresponding taxa were found through a BLAST search, as later referenced in line 99. 

As requested by reviewer#2, we added the taxon ID: NCBI:txid2509509 (line 64).  Note that there is no taxon ID in the ICTV database.

In the Discussion section 4.2 (SADS-CoV evolved from Rhinolophus affinis circulating in Guangdong), line 645, the authors write: "Our phylogeographic analyses based on either 95 or 87 Rhinacovirus genomes [...]". I understand that eight problematic sequences have been removed from the initial dataset, as described in Section 3.7. However, I suggest either removing "or 87" or repeating in the Discussion the reason that excluded those eight sequences.

Changed as requested by reviewer#2 as follows (line 679): “Our phylogeographic analyses of based on either 95 or 87 Rhinacovirus genomes…”

Reviewer 3 Report

Comments and Suggestions for Authors

1.        Figure S1-S5, the bootstrap values are full displayed, please adjust.

Author Response

Comments and Suggestions for Authors

  1. Figure S1-S5, the bootstrap values are full displayed, please adjust.

Thanks for the comment.

The five figures S1-S5 were modified as requested by reviewer#3.

Reviewer 4 Report

Comments and Suggestions for Authors

The manuscript authored by Hassanin et al. reports the discovery of five novel rhinacovirus genomic sequences from horseshoe bat samples collected in Vietnam. Through precise phylogenetic analyses, the authors have identified that these rhinacoviruses can be clustered into four different groups. Furthermore, detailed phylogeographic investigations have revealed that these five viruses found in Vietnam show close genetic relationships with previously documented viruses found in neighboring regions, except Rt22QT46R. The analyses further support the direct origin of SADS-CoV might be from Rhinolophus affinis in Guangdong. Besides, some debatable sequences were raised after sequence alignment and recombination analyses. This paper provides new insight into the origin and the evolutionary route of rhinacovirus. Overall, this manuscript is suitable for publishing in Viruses. There are some comments below for author’s consideration:

1.     The discovery of potentially problematic genomes by authors should be encouraged. However, the analyses may not provide sufficient evidence to support the claim of "Evidence for several problematic genomes." Authors should revise the statement in this section to incorporate possibility tones.

2.     There seems more GRPS in SuperTRI consensus tree (Fig 3) than that in Maximum-likelihood tree (Fig 4/5). Which tree is more presentative clustering of these sequences?

3.     Rt22QT46R (clustered in SADSCoVr) was found in the middle of Vietnam, which is far away from GD and GX. Besides, the viral reads number was much higher than other samples. Is it possible that SADS-CoV or SADSCoVr have already circulated in this area? Readers may have more interest in this strain, authors may provide some more detailed analyses and description in the main text.

4.     The concluding statement in the abstract (line29-31) is somewhat misleading, the virus has been spreading silently in pig farms at least seven years before the first SADS outbreak, or at least seven years from now?

5.     The citation 11 in line 69 is not suitable, unless it has been accepted or published in preprint platform.

6.     The calculation methods of phylogenetic contributions should be thoroughly described, so that readers can better understand the contributions of the different parameters.

7.     Statistical methods should be provided.

8.     Feedback to section 3.6 “On the geographic sampling of Ra160660”. I have inquired the detailed information of Ra160660 from the authors. This sample was collected from R.a. in Yunnan province, China, 2016. authors can add more discussion about your phylogeographic investigations.

Author Response

The manuscript authored by Hassanin et al. reports the discovery of five novel rhinacovirus genomic sequences from horseshoe bat samples collected in Vietnam. Through precise phylogenetic analyses, the authors have identified that these rhinacoviruses can be clustered into four different groups. Furthermore, detailed phylogeographic investigations have revealed that these five viruses found in Vietnam show close genetic relationships with previously documented viruses found in neighboring regions, except Rt22QT46R. The analyses further support the direct origin of SADS-CoV might be from Rhinolophus affinis in Guangdong. Besides, some debatable sequences were raised after sequence alignment and recombination analyses. This paper provides new insight into the origin and the evolutionary route of rhinacovirus. Overall, this manuscript is suitable for publishing in Viruses. 

Thanks for the comments

There are some comments below for author’s consideration:

  1. The discovery of potentially problematic genomes by authors should be encouraged. However, the analyses may not provide sufficient evidence to support the claim of "Evidence for several problematic genomes." Authors should revise the statement in this section to incorporate possibility tones.

As requested by reviewer#4, we changed the title of the paragraph as follows (line 395): “3.4. Evidence for Detection of several problematic genomes”

  1. There seems more GRPS in SuperTRI consensus tree (Fig 3) than that in Maximum-likelihood tree (Fig 4/5). Which tree is more presentative clustering of these sequences?

In Fig. 4 and 5, the GB barcodes (with GRPS) are shown only for a selection of nodes in conflict with the supertree consensus of Fig. 3. This explains why there are less GB barcodes than in Figure 3. In addition, all internal branches marked by a red zero were found supported by no genomic fragments (GRPS = 0%). Since this point was not specified in the legend of Figure 3, we included a new sentence as follows (lines 344-345): Internal branches marked by a red zero were not supported by any genomic fragment (GRPS = 0%).

  1. Rt22QT46R (clustered in SADSCoVr) was found in the middle of Vietnam, which is far away from GD and GX. Besides, the viral reads number was much higher than other samples. Is it possible that SADS-CoV or SADSCoVr have already circulated in this area? Readers may have more interest in this strain, authors may provide some more detailed analyses and description in the main text.

To our knowledge, there were some pig diarrhea outbreaks in Vietnam during recent decades but they were caused by PED and African swine fever viruses. However, we agree with reviewer #4 that SADSCoVr might have already circulated in pigs of Vietnam. Additional studies are needed to test the hypothesis.

  1. The concluding statement in the abstract (line29-31) is somewhat misleading, the virus has been spreading silently in pig farms at least seven years before the first SADS outbreak, or at least seven years from now?

Corrected as follows (lines 29-32):  “The reliable data currently available therefore suggests that all SADS-CoV strains originate from a single bat source and that the virus has been spreading silently in pig farms in several provinces of China for at least seven years after the first outbreak in August 2016.”

  1. The citation 11 in line 69 is not suitable, unless it has been accepted or published in preprint platform.

Corrected.  The preprint is now fully referenced.

  1. The calculation methods of phylogenetic contributions should be thoroughly described, so that readers can better understand the contributions of the different parameters.

As requested by reviewer#4, we added several sentences to better describe the calculation methods of phylogenetic contributions as follows (lines 216-227): “The geographic CGB barcodes (showing the geographic origins of the closely-related viruses) were derived from the original phylogenetic CGB barcodes by choosing different colours for viruses collected in distinct geographic areas.

Following [21], the CGB barcodes were used to calculate the phylogenetic contributions of viruses (CT), geographic areas (CTG), and host species (CTH). The exclusive contributions were also calculated for viruses (CE), geographic areas (CEG), and host species (CEH) using only bipartitions showing exclusive ancestry with the virus of interest. For instance, the total geographic contribution (CTG) of Yunnan in the CGB barcode of Ra160660 was calculated by summing the GRPS intervals in the whole-genome alignment in which Ra160660 was found closely-related to one or several viruses from Yunnan and other geographic areas. Then, the sum was multiplied by 100 and divided by the total length of our alignment (27,250 nt) to obtain the percentage contribution. Similarly, the exclusive geographic contribution (CEG) of Yunnan was calculated by summing the GRPS intervals in which Ra160660 shared close relationships with one or more viruses from Yunnan only.

The first sentence was removed because we did not show any geographic CGB barcode in the figures presented in this study.

  1. Statistical methods should be provided.

The statistical method is now indicated in section 2.6 (lines 227-229):

McNemar’s Chi-squared tests were used to perform pairwise comparison between the highest geographic contributions, as well as between the highest host contributions.

  1. Feedback to section 3.6 “On the geographic sampling of Ra160660”. I have inquired the detailed information of Ra160660 from the authors. This sample was collected from R.a. in Yunnan province, China, 2016. Authors can add more discussion about your phylogeographic investigations.

Thanks for the information. We added the following sentence at the end of section 3.6 (lines 599-601):Our prediction was confirmed by one of our reviewers, who informed us that Ra160660 was collected in Yunnan in 2016.”